# The Effect of Muicle–Chitosan Edible Coatings on the Physical, Chemical, and Microbiological Quality of Cazon Fish (*Mustelus lunulatus*) Fillets Stored in Ice

**DOI:** 10.3390/foods14091619

**Published:** 2025-05-03

**Authors:** José Alberto Cruz-Guzmán, Alba Mery Garzón-García, Saúl Ruíz-Cruz, Enrique Márquez-Ríos, Santiago Valdez-Hurtado, Gerardo Trinidad Paredes-Quijada, José Carlos Rodríguez-Figueroa, María Irene Silvas-García, Nathaly Montoya-Camacho, Victor Manuel Ocaño-Higuera, Dalila Fernanda Canizales-Rodríguez, Edgar Iván Jiménez-Ruíz

**Affiliations:** 1Departamento de Investigación y Posgrado en Alimentos, Universidad de Sonora, Blvd, Luis Encinas y Rosales s/n, Hermosillo 83000, Mexico; a216201141@unison.mx (J.A.C.-G.); saul.ruizcruz@unison.mx (S.R.-C.); enrique.marquez@unison.mx (E.M.-R.); maria.silvas@unison.mx (M.I.S.-G.); 2Ingeniería Industrial, Universidad del Valle, Sede Regional Caicedonia, Carrera 14 No 4-48, Caicedonia 762540, Valle del Cauca, Colombia; garzon.alba@correounivalle.edu.co; 3Unidad Académica Navojoa, Universidad Estatal de Sonora, Blvd, Manlio Fabio Beltrones 810, Col. Bugambilias, Navojoa 85875, Mexico; santiago.valdez@ues.mx; 4Departamento de Ciencias Químico Biológicas, Universidad de Sonora, Blvd. Luis Encinas y Rosales s/n, Hermosillo 83000, Mexico; gerardo.paredes@unison.mx (G.T.P.-Q.); nathaly.montoya@unison.mx (N.M.-C.); 5Departamento de Ingeniería Química y Metalurgia, Universidad de Sonora, Hermosillo 83000, Mexico; jose.rodriguez@unison.mx; 6Unidad de Tecnología de Alimentos, Secretaría de Investigación y Posgrado, Universidad Autónoma de Nayarit, Ciudad de la Cultura s/n, Tepic 63000, Mexico; edgar.jimenez@uan.edu.mx s

**Keywords:** *Justiciera spicigera*, edible coating, cazon fish, quality, antibacterial activity

## Abstract

Fishery products are highly perishable; therefore, effective preservation strategies are essential to maintain their freshness, quality, and shelf life. One promising approach involves the use of edible coatings formulated with natural extracts, such as muicle (*Justicia spicigera*). This study evaluated the effect of a muicle–chitosan edible coating on the physical, chemical, and microbiological quality of cazon fish (*Mustelus lunulatus*) fillets stored in ice for 18 days. The muicle extract was obtained by macerating dried leaves for 48 h, and its antibacterial activity was subsequently assessed. A control group (C) and three treatments—muicle extract (ME), chitosan (CH), and a combined muicle–chitosan coating (MECH)—were applied and monitored throughout the storage period. Quality parameters, including pH, colour, water-holding capacity (WHC), texture, total volatile basic nitrogen (TVB-N), and mesophilic microbial counts, were evaluated. The muicle extract exhibited antibacterial activity, with MIC and IC_50_ values of 3.01 ± 0.73 and 204.56 ± 20.23 µg/mL against *Shewanella putrefaciens*, and 0.10 ± 0.07 and 118.09 ± 14.51 µg/mL against *Listeria monocytogenes*, respectively. Treatments of ME, CH, and MECH significantly improved (*p* < 0.05) the quality of fillets by reducing TVB-N, pH, and microbial load compared to the control. In conclusion, the muicle extract demonstrated antibacterial potential and, either alone or in combination with chitosan, effectively preserved the physical, chemical, and microbiological quality of cazon fillets during ice storage.

## 1. Introduction

Fishery products are among the most perishable food commodities, primarily due to postmortem physical, chemical, and microbiological changes in muscle tissue. These changes are closely associated with the fish’s chemical composition, muscle structure, enzymatic activity, microbial load, and both capture and post-capture handling conditions [1]. It is well established that immediately after death, fish undergo a rapid decline in freshness and quality, initially driven by endogenous enzymatic reactions and subsequently by microbial enzymatic activity [2]. Under optimal harvesting and post-harvest conditions, the shelf life of fish stored in ice typically ranges from 15 to 18 days; however, improper handling practices can significantly shorten this period by accelerating spoilage through temperature abuse and enhanced microbial growth [1,3].

One species of economic importance in Mexico is cazon (*Mustelus lunulatus*), with a reported production of 9808 tons in 2022 [4]. Belonging to the Triakidae family and classified as an elasmobranch, cazon resembles small sharks, reaching lengths of approximately 125 cm and lifespans of up to 10 years [5]. This species is found in the Gulf of California and along the Pacific coast of Mexico [6]. It is typically caught by artisanal fishers using small boats with outboard motors [7], often under suboptimal harvesting and slaughter conditions. Such inadequate handling practices significantly compromise freshness and quality, negatively affecting the fillet’s physical, chemical, microbiological, and sensory attributes [1,8] and ultimately posing health risks to consumers and causing economic losses for both producers and processors.

Conventional preservation methods include moisture reduction (drying and salting), temperature control (refrigeration and freezing), and heat treatments (pasteurization, sterilization, and smoking, canning). However, recent advancements have focused on innovative alternatives aimed at preserving freshness and extending shelf life, such as high-pressure processing [9], pulsed electric fields [10], and active or intelligent packaging systems [11]. Of particular interest are natural extracts and essential oils derived from herbs, spices, and plants [12,13,14,15,16], which exhibit antimicrobial and antioxidant properties. These compounds can be applied independently or incorporated into polymer-based edible coatings that serve as physical and biochemical barriers against spoilage [17].

Recent studies have demonstrated the effectiveness of edible coatings in preserving fish quality. For instance, Ramírez-Guerra et al. [18] reported that an edible coating composed of tomato plant extract and chitosan extended the shelf life of sierra (*Scomberomorus sierra*) fillets stored on ice for six days. Similarly, Xiong et al. [19] showed that a coating formulated with salmon bone gelatin, chitosan, gallic acid, and clove oil extended the shelf life of salmon fillets by five days at 4 °C. In another study, Lee et al. [20] found that chitosan–ascorbic acid coating prolonged the shelf life of tilapia fillets by nine days compared to the controls. These findings underscore the potential of natural polymer-based coatings as effective strategies for maintaining quality and extending the shelf life of fishery products.

Among the various plant sources of bioactive compounds, muicle (*Justicia spicigera*) has gained attention due to its reported medicinal properties, including antidiabetic, anti-inflammatory, antiproliferative, antiurolithic, neuroprotective, and hypolipidemic effects [21]. The leaves and flowers of muicle, particularly in ethanolic extracts, are rich in phenolic compounds that exhibit antioxidant, antimicrobial, and antitumor activities [22]. These extracts have demonstrated efficacy against *Shigella flexneri*, *Salmonella typhi*, *Salmonella typhimurium*, *Escherichia coli*, and *Staphylococcus aureus* [21]. Chitosan, a biopolymer primarily derived from crustacean shells via the alkaline deacetylation of chitin [23], is also widely recognized for its antimicrobial and antioxidant properties [24]. Therefore, the aim of this study was to evaluate the effect of an edible coating composed of aqueous muicle extract and chitosan on the physical, chemical, and microbiological quality of cazon fillets during 18 days of ice storage.

## 2. Materials and Methods

### 2.1. Specimen Collection

Cazon fish specimens were obtained from artisanal fishermen operating in the Gulf of California, near the coast of Bahía de Kino, Sonora, Mexico. The freshly caught fish, with an average weight of 0.7 kg, were immediately stored in airtight coolers with alternating layers of fish and ice and transported to the Food Research Laboratory (LIA) at the Department of Chemical-Biological Sciences, University of Sonora, in Hermosillo, Sonora, Mexico. The time between capture and arrival at the laboratory did not exceed six hours. Upon arrival, the fish were filleted and rinsed with a chilled mixture of distilled water and ice. The fillets were then randomly assigned to four experimental groups: a control group (C; no treatment) and three treatment groups—the muicle extract (ME), chitosan (CH), and muicle–chitosan combination (MECH) groups.

### 2.2. Extract Preparation

Muicle extract was prepared following the method described by Burkhard et al. [25]. Muicle plants were acquired in Hermosillo, Sonora, Mexico, and cultivated for four months to allow for leaf growth. The harvested leaves were washed and dehydrated using a convection oven (Thermo Scientific, San Luis Potosí, Mexico) at 45 °C for 8 h. A total of 20 g of dried, crushed leaves were placed into a 1 L Erlenmeyer flask containing 400 mL of distilled water at 60 °C and were allowed to macerate for 48 h at room temperature under dark conditions. The resulting mixture was filtered using Whatman No. 4 paper and a Büchner funnel under vacuum. The filtrate was lyophilized in a freeze dryer (Labconco Corporation, Kansas City, MO, USA) for four days and stored at −20 °C (Torrey, Zapopan, Jalisco, Mexico) until use.

### 2.3. Evaluation of Antibacterial Activity

#### 2.3.1. Bacterial Activation and Culture Preparation

The antibacterial activity of the muicle extract was evaluated against *S. putrefaciens* (ATCC 8071) and *L. monocytogenes* (ATCC 181115) following the method described by Esparza-Espinoza et al. [26], with slight modifications. Bacterial inocula were prepared by transferring colonies from pure cultures into Mueller–Hinton broth (pH 7.2) and incubating them at 35–37 °C for 18–24 h in a Thermo Electron incubator (LED GmbH, Langenselbold, Germany). The optical density (OD) of the cultures was measured at 600 nm using a double-beam spectrophotometer (Agilent Technologies, Penang, Malaysia) and adjusted to a 0.5 McFarland standard, corresponding to approximately 10^8^ CFU/mL. A 1.58 mL aliquot of the adjusted bacterial suspension was subsequently diluted with Mueller–Hinton broth to achieve a final concentration of approximately 10^6^ CFU/mL for antibacterial testing.

#### 2.3.2. Determination of Minimum Inhibitory Concentration (MIC) and IC_50_

MIC values were determined according to Esparza-Espinoza et al. [26]. Five concentrations of muicle extract (14–230 μg/mL) were prepared in distilled water and diluted 1:10 with BHI broth, adjusting the bacterial concentration to 10^6^ CFU/mL. A 96-well microplate was loaded with 150 μL of the extract and 50 μL of bacterial suspension (10^8^ CFU/mL). After 24 h of incubation at 37 °C, absorbance was read using a Multiskan GO microplate reader (Thermo Scientific, New York, NY, USA). MIC and IC_50_ values were calculated using probit regression analysis in NCSS version 24.0.1.

### 2.4. Determination of Phytochemical Compounds

#### 2.4.1. Total Phenolic Content

The total phenolic content was determined using the Folin–Ciocalteu method described by Silva-Beltrán et al. [27], with slight modifications. In a microplate, 10 µL of extract was combined with 25 µL of the Folin–Ciocalteu reagent (1 N), 25 µL of 20% Na_2_CO_3_, and 140 µL of distilled water. After 30 min in the dark, absorbance was measured at 760 nm using a Multiskan GO microplate reader. The results were expressed in mg gallic acid equivalents (mg GAE).

#### 2.4.2. Total Flavonoid Content

Total flavonoids were quantified according to Silva-Beltrán et al. [27], with modifications. In a microplate, 80 µL of the muicle extract was mixed with 80 µL of 2% aluminum trichloride solution. The mixture was incubated for 60 min in darkness, and absorbance was recorded at 415 nm. The results were expressed in mg quercetin equivalents (mg QE).

### 2.5. Preparation and Application of Treatments and Control

Chitosan solution (CH) was prepared by dissolving 1% (*w*/*v*) chitosan in 1% (*v*/*v*) acetic acid containing 0.5% (*v*/*v*) glycerol as the plasticizer. The mixture was homogenized at 36,000 rpm for 3 min. For MECH (edible coating), the lyophilized muicle extract was added to the chitosan solution at 0.3% (*w*/*v*). The ME solution was prepared by dissolving muicle extract in water at 0.3% (*w*/*v*). The treatments described above were compared with a control (C) (without CH or ME). Cazon fillets were immersed in each treatment for 2 min, drained for 5 min, and then individually sealed in high-density polyethylene bags for further analysis.

### 2.6. Ice Storage Study

Fillets from the C, ME, CH, and MECH groups were stored in hermetically sealed coolers with crushed ice for 18 days. Ice was replaced as needed. Samples were collected on days 0, 3, 6, 9, 12, 15, and 18. Evaluations included shear force, pH, colour, WHC, TVB-N, and aerobic mesophilic counts. All analyses were conducted in quadruplicate per sampling point.

### 2.7. Analytical Procedures

#### 2.7.1. pH

pH was determined following the method of Woyewoda et al. [28]. A 2 g sample of fish muscle was homogenized with 18 mL of distilled water using an Ultra-Turrax T18 Basic homogenizer (IKA Works Inc., Wilmington, NC, USA) at 18,000 rpm for 1 min. pH was measured using a Hanna Instruments HI97107 pH meter calibrated with standard buffers.

#### 2.7.2. Colour

Colour was measured using a MiniScan HunterLab colorimeter (Preston, VA, USA) operating in reflectance mode with a 0.5 cm aperture. L* (lightness), a* (red–green), and b* (yellow–blue) values were recorded from the dorsal region of the fillet.

#### 2.7.3. Texture

Texture was assessed according to Canizales-Rodríguez [29] using a Shimadzu texture analyzer (Model EZS 346-54909-33) fitted with a Warner–Bratzler shear cell. Fillet samples (10 × 10 × 20 mm) were oriented parallel to the muscle fibres, and maximum shear force (N) was recorded with a 50 kg load cell at a speed of 100 cm/min.

#### 2.7.4. Water-Holding Capacity (WHC)

WHC was measured following the method of Cheng et al. [30]. Two grams of the sample was centrifuged at 7500× *g* for 30 min at 4 °C using a Thermo IEC MULTI-RF centrifuge (Thermo Fisher Scientific, Asheville, NC, USA). WHC was calculated based on water loss relative to the initial sample weight.

#### 2.7.5. Total Volatile Basic Nitrogen (TVB-N)

TVB-N was determined using the protocol of Woyewoda et al. [28]. Two grams of muscle was blended with 2 g of magnesium oxide and 300 mL of distilled water. The mixture was homogenized with an Ultra-Turrax T18 (Ika Works Inc., Wilmington, NC, USA) and defoamed with 20 drops of vegetable oil. Volatile bases were distilled for 25 min and captured in 15 mL of 2% boric acid, then titrated with 0.05 N sulfuric acid. A blank was processed identically. The results were expressed as mg nitrogen per 100 g of the sample.

#### 2.7.6. Aerobic Mesophilic Count

The microbial load was determined according to the Mexican Official Standard [31] for aerobic plate counts. Ten grams of the sample was blended with 90 mL of sterile 1% peptone water. Five serial dilutions were prepared, and 1 mL from each was plated in triplicate on Plate Count Agar (Oxoid Ltd., Basingstoke, UK). Plates were incubated at 35 ± 2 °C for 48 h, and the results were expressed as colony-forming units per gram (CFU/g).

### 2.8. Statistical Analysis

Data were analyzed using a completely randomized design. Means and standard deviations were calculated. Antibacterial activity, phytochemical content with antioxidant potential in the muicle extract, and microbiological analyses were each conducted in duplicate. In contrast, all other analytical determinations performed during the ice storage period were carried out in quadruplicate, with four replicates per sampling day and per analytical method. One-way ANOVA was applied, and significant differences were analyzed using Tukey’s test with a 5% significance level (*p* < 0.05) using NCSS software version 6.0.22 (NCSS, Statistical Sofware, Kaysville, UT, USA).

## 3. Results

### 3.1. Evaluation of Antibacterial Activity and Phytochemical Compounds with Antioxidant Potential in Muicle Extract

#### 3.1.1. Minimum Inhibitory Concentration (MIC) and IC_50_ of Muicle Extract

MIC is a widely used indicator to assess the effectiveness of antibacterial compounds, as it defines the minimum concentration required to inhibit bacterial growth and proliferation. In this study, the MIC of an aqueous extract of muicle was evaluated against *S. putrefaciens* and *L. monocytogenes*, which are two bacteria associated with the deterioration of fish product quality.

Figure 1 shows the inhibitory effect of various concentrations of the muicle aqueous extract on the growth of *S. putrefaciens* and *L. monocytogenes*. At the highest concentration tested (230 µg/mL), inhibition rates of 53.82 ± 5.87% and 57.71 ± 3.35% were observed for *S. putrefaciens* and *L. monocytogenes*, respectively. At the lowest concentration (14 µg/mL), inhibition rates decreased to 11.72 ± 2.03% and 23.77 ± 2.61%, respectively. These results indicate that muicle extract is more effective against *L. monocytogenes*. This observation is consistent with previous reports describing the antibacterial activity of the muicle plant, which is attributed to the presence of phenolic compounds and flavonoids. These phytochemicals are known to disrupt bacterial cell membranes and interfere with key metabolic pathways, ultimately inhibiting bacterial growth and survival [32].

Additionally, the MIC (µg/mL) and IC_50_ values of the aqueous muicle extract against *S. putrefaciens* and *L. monocytogenes* are presented in Table 1. The minimum inhibitory concentrations were 3.01 ± 0.73 µg/mL for *S. putrefaciens* and 0.09 ± 0.09 µg/mL for *L. monocytogenes*. In terms of IC_50_ values, 204.56 ± 20.22 µg/mL and 118.09 ± 14.51 µg/mL were required to inhibit 50% of *S. putrefaciens* and *L. monocytogenes* populations, respectively. These findings underscore the antimicrobial potential of the muicle extract, the efficacy of which varies depending on both the concentration applied and the bacterial species targeted. It is worth noting that, in this study, the muicle extract was incorporated into an edible coating formulation in combination with chitosan—a natural biopolymer extensively studied for its antimicrobial properties. The synergistic use of chitosan with natural plant extracts offers a safe and biodegradable strategy for the preservation of fishery products [33]. Thus, the results suggest that muicle extract holds significant promise as an active ingredient in edible coatings for fish and seafood. Its demonstrated ability to inhibit spoilage-related bacteria such as *S. putrefaciens* and *Listeria monocytogenes* —key contributors to quality loss—positions it as a potential alternative to extend shelf life and maintain the quality of fishery products.

#### 3.1.2. Total Phenolic and Flavonoid Content in Muicle Extract

Phenols and flavonoids are a broad class of phytochemicals structurally characterized by aromatic rings bearing hydroxyl groups. These compounds act as natural antioxidants, counteracting oxidative stress and cellular damage and thereby reducing the risk of diseases associated with these conditions [34].

Table 2 presents the total phenolic content in the aqueous extract of muicle. The results show a concentration of phenolic compounds of 41.48 ± 3.28 mg GAE/100 g for an extract concentration of 230 μg/mL. This value was lower than that reported by Alvarez-Poblano et al. [35], who obtained 89.36 ± 0.76 mg GAE/100 g in their aqueous extract of muicle. Variations in phenolic content between studies can be attributed to several factors, including soil composition during plant cultivation, the timing of leaf harvesting, the drying method applied, and differences in extract concentration and preparation procedures. These parameters significantly influence the efficiency and yield of phenolic compound extraction.

On the other hand, the flavonoid content in the extract of muicle was found to be 27.52 ± 1.11 mg QE/100 g of the sample at an extract concentration of 230 µg/mL. The presence of flavonoids and phenolic compounds—such as kaempferitrin and kaempferol trirhamnoside, known for their bioactive properties—suggests a high antioxidant potential in muicle extract [36]. These findings support its potential use as a protective agent for preserving the quality of fishery products.

Phenolic and flavonoid compounds possess antioxidant properties that enhance food stability by delaying oxidative deterioration. They may also provide health benefits by reducing oxidative stress associated with chronic diseases [37]. This functionality supports the potential application of muicle extract, either alone or combined with other compounds, in developing edible coatings for fish products. Edible coatings enriched with bioactive compounds have been shown to act as effective barriers against oxidation and microbial spoilage. Therefore, incorporating muicle extract into edible coating systems represents a promising and sustainable strategy to improve the shelf life and quality of fishery products.

### 3.2. Evaluation of Physical Quality

#### 3.2.1. pH Determination

Muscle pH is a key indicator of the freshness and overall quality of fishery products [38]. Figure 2 illustrates the pH evolution in cazon fillets from the control group (C) and the treated groups (ME, CH, and MECH) during 18 days of ice storage. The initial pH in the control group was 6.80 ± 0.08, which is higher than the value reported by Ocaño-Higuera et al. [39] for cazon muscle (6.43) and also higher than the value of 6.48 reported by Tomé et al. [40] for tilapia (*Oreochromis* spp.) stored on ice.

Differences in initial pH may be attributed to factors such as species, season, fish size, stress levels prior to and during capture, and diet composition [25,39,41]. In the treated groups, initial pH values were 6.63 ± 0.20 (ME), 6.73 ± 0.17 (CH), and 6.63 ± 0.09 (MECH). No significant differences (*p* < 0.05) were observed between these values and the control group. The relatively lower initial pH values may be associated with the production and accumulation of lactic acid in the muscle, which is a compound generated through glycogen degradation following capture [39].

During ice storage, the pH of the control group (C) increased significantly (*p* < 0.05), reaching a final value of 7.4 ± 0.75. This increase is commonly associated with a loss of freshness and quality and may result from the accumulation of ammonia and other amines—alkaline compounds that raise muscle pH levels [42]. Volatile amines such as histamine, cadaverine, and putrescine can be produced via autolytic degradation or through bacterial activity, particularly by *S. putrefaciens* acting on free amino acids [43]. In contrast, pH values in the ME, CH, and MECH treatments remained relatively stable throughout storage, with final values of 6.65 ± 0.50, 6.50 ± 0.00, and 6.52 ± 0.05, respectively. This stabilization may be attributed to the antimicrobial and antioxidant properties of muicle extract and chitosan, either individually or in combination. The presence of phenolic compounds in the muicle extract, such as kaempferitrin and kaempferol trirhamnoside, may contribute to its bioactivity in preserving the quality of fishery products [21]. Meanwhile, the antimicrobial effectiveness of chitosan is known to be influenced by factors including molecular weight, concentration, pH, origin, and temperature [44]. These properties support the potential of muicle- and chitosan-based coatings as a viable preservation strategy for fish products.

#### 3.2.2. Texture

Texture is one of the most critical attributes for evaluating the quality of fishery products [45,46]. In this study, texture was assessed through a shear force test. Figure 3 displays the shear force results for cazon fillets from the control group (C) and the treated groups (ME, CH, and MECH) during 18 days of ice storage. The initial shear force in the control group was 44.86 ± 1.08 N, which is lower than the 70.60 ± 1.22 N reported by Ocaño-Higuera et al. [39] for cazón. These differences can be attributed to several factors, including species, age, sex, genotype, nutrition, and environmental conditions [47].

No significant differences (*p* > 0.05) were found between the control and treatment groups at the beginning of storage, with initial shear force values of 40.22 ± 3.10 N (ME), 40.46 ± 2.32 N (CH), and 42.19 ± 4.89 N (MECH). However, a significant decrease (*p* < 0.05) in shear force was observed in all groups over time. The final values were 26.81 ± 3.38 N (C), 35.50 ± 3.18 N (ME), 31.51 ± 5.73 N (CH), and 27.36 ± 3.29 N (MECH). No significant differences were observed among treatments at the end of the storage period (*p* > 0.05). Similar texture degradation was reported by Chaparro-Hernández et al. [48] in tilapia (*Oreochromis niloticus*) treated with chitosan–carvacrol edible coatings during 21 days of ice storage. Muscle firmness in fish is closely linked to collagen content, which is a major structural component of muscle tissue [49]. Softening observed in both control and treated fillets during storage may be associated with factors such as Z-line degradation and collagen denaturation. In addition, endogenous protease activity—specifically the degradation of myofibrillar proteins—is a key factor in postmortem softening. These proteases exhibit optimal activity at pH values ranging from 5.5 to 6.5 [50].

#### 3.2.3. Water-Holding Capacity (WHC)

Water-holding capacity (WHC) is one of the most widely used parameters for assessing the quality of fishery products [51]. This indicator refers to the muscle’s ability to retain water within its fibres and protein structures when subjected to external forces such as compression or centrifugation under defined conditions [52].

Figure 4 illustrates the WHC behaviour of cazon fillets from the control group (C) and the treatment groups (ME, CH, and MECH) during 18 days of ice storage. Initial WHC values were 89.83 ± 2.01% for the control group and 89.12 ± 1.42%, 87.60 ± 8.28%, and 86.75 ± 0.94% for the ME, CH, and MECH treatments, respectively. No significant differences were observed among the groups (*p* > 0.05). These initial values were lower than those reported by Ocaño-Higuera et al. [39], who recorded an initial WHC of 92.7 ± 1.9% in the cazon muscle. However, they were notably higher than the initial WHC of 52.43 ± 0.03% reported by Wijesundara et al. [53] for the elasmobranch *Dasyatis margarita* (Thornback ray). Such variations in WHC are commonly attributed to factors such as pH, rigour mortis stage, temperature, species, stress level, and slaughter conditions [54].

Furthermore, no significant differences (*p* > 0.05) were observed in the WHC either over time or among treatments throughout the storage period. The final WHC values were 89.36 ± 1.53% (control), 86.15 ± 1.12% (ME), 87.87 ± 1.15% (CH), and 88.12 ± 0.35% (MECH). These values are consistent with the findings of Ocaño-Higuera et al. [39], who reported a final WHC of 86.2% in cazon muscle after 18 days of storage in ice.

#### 3.2.4. Colour

Colour is a key quality parameter in fishery products, as it directly influences consumer perception and acceptance [55]. In this study, colour was evaluated using the L* (lightness), a* (red–green), and b* (yellow–blue) coordinates. Figure 5a presents the evolution of the L* parameter in cazon fillets from the control group (C) and the ME, CH, and MECH treatments during 18 days of ice storage. The fillets treated with CH showed slightly higher L values, although no significant differences were observed among treatments (*p* > 0.05), which were comparable to those reported by Montaño-Cota et al. [56] for sierra (*Scomberomorus sierra*) fillets. However, the CH treatment showed the highest L* value, indicating greater lightness, brightness, and transparency—attributes associated with reduced darkness. This trend is consistent with the findings of Da Silva Santos et al. [57] in smoked tilapia fillets coated with chitosan, which may be explained by the acidic nature of the chitosan solution. The low pH of the coating can cause the slight coagulation of surface proteins, promoting the leaching of muscle pigments [58]. Regarding the effect of storage time within treatments, the CH group exhibited a significant increase in the L value (*p* < 0.05), reaching a final value of 63.27 ± 5.13. This increase in lightness may be attributed to exudation and fluid loss, resulting in a more watery and reflective surface appearance on the fillet [3]. However, when comparing treatments at the end of the storage period, no significant differences (*p* > 0.05) were observed between CH-treated fillets and control fillets.

Figure 5b shows the behaviour of the a* parameter. Initial values were 12.75 ± 3.68 (C), 12.63 ± 2.94 (ME), 12.63 ± 3.68 (CH), and 15.29 ± 2.94 (MECH). These values are higher than the value of 7.45 reported by Montaño-Cota et al. [56] in the sierra muscle. Throughout storage, a significant decrease (*p* < 0.05) in a* values was observed across all groups, ending at 5.11 ± 0.71 (C), 2.29 ± 1.33 (ME), 7.11 ± 3.95 (CH), and 7.03 ± 1.27 (MECH). The ME treatment showed the lowest final a* value, indicating a marked loss in red coloration, which may be attributed to the oxidation of sarcoplasmic proteins such as myoglobin (converted to metmyoglobin) and hemoglobin, which can be solubilized in the water released from the muscle during storage [59].

Figure 5c illustrates the changes in b* values of cazon fillets over 18 days of ice storage. A consistent decreasing trend was observed across all treatments, with significant differences over time (*p* < 0.05) and between treatments (*p* < 0.05). This reduction in b* values indicates a gradual loss of yellowness in the fillets, which is likely associated with oxidative processes and pigment degradation during cold storage, as previously reported for fishery products [60]. Among the treatments, fillets coated with muicle extract (ME) exhibited a slightly greater decrease in b* values; however, the inclusion of muicle still contributed to the preservation of colour attributes throughout storage. These results highlight the potential of muicle extract as a natural additive for maintaining visual quality and extending the shelf life of fish fillets under ice storage conditions.

### 3.3. Chemical Determinations

#### Total Volatile Basic Nitrogen (TVB-N)

The determination of total volatile basic nitrogen (TVB-N) is one of the most widely used methods for evaluating the spoilage of fishery products, as it reflects the accumulation of nitrogenous volatile compounds produced through microbial and enzymatic activity [61,62]. Figure 6 shows the changes in TVB-N values of cazon fillets from the control group (C) and the ME, CH, and MECH treatments during 18 days of ice storage.

Initial TVB-N values ranged between 26.05 ± 2.64 and 29.40 ± 1.84 mg N/100 g, with no significant differences among the control and treated groups (*p* > 0.05). These values were lower than those reported by Ocaño-Higuera et al. [39], who recorded 40.0 ± 3.3 mg N/100 g in fresh cazon fillets. Moreover, the values observed in this study were below the maximum permissible limit of 35 mg N/100 g established by both Mexican and international regulations for fishery products deemed suitable for human consumption [63,64].

During ice storage, TVB-N levels increased significantly (*p* < 0.05) in all groups. The final values were 68.62 ± 3.42 (C), 43.98 ± 2.81 (ME), 48.86 ± 1.26 (CH), and 46.41 ± 2.33 mg N/100 g (MECH). Statistically significant differences (*p* < 0.05) were observed between the control and all treated groups. The greatest increase in TVB-N was observed in the control group, which did not receive any preservation treatment, resulting in greater deterioration compared to the ME, CH, and MECH treatments. These increases are consistent with previous reports in elasmobranchs [39,65], where TVB-N accumulation is primarily attributed to ammonia and other amines—such as NH_3_—resulting from the decomposition of nitrogenous compounds like urea present in the muscle tissue [1].

According to the cited regulations, the limit of 35 mg of TVB-N/100 g defines the threshold for human consumption. In this study, the control group exceeded this limit by day 9 of storage. In contrast, the CH and MECH treatments exceeded this limit on day 12, while the ME treatment only exceeded the limit on day 15. These findings demonstrate the effectiveness of ME and CH, alone or combined, as edible coatings for extending the shelf life of cazon fillets. This conclusion is supported by a complementary (non-reported) sensory evaluation, in which the control group exhibited off-odours by day 9, though they were not yet putrid.

It is important to note that all fish and fillets were handled under strict hygienic conditions and stored under refrigeration (on ice), which helped minimize metabolic, chemical, and enzymatic deterioration, thus maintaining acceptable quality during the storage period. Nonetheless, elevated TVB-N values are a common characteristic of elasmobranchs due to their high endogenous levels of nitrogenous compounds, particularly urea. In species like sharks and rays, urea concentrations can reach up to 2000 mg/100 g of muscle [1], and its breakdown into ammonia contributes to elevated TVB-N values, distinguishing them from teleost fish.

It is also relevant to consider that in countries like Peru, the National Fishery Health Agency (SANIPES) established a maximum permissible limit of 70 mg N/100 g for cazon fillets [66]. This highlights the need for further research to establish species-specific TVB-N thresholds for elasmobranchs such as cazon and rays, which may differ from those applied to teleost species. Finally, the lower final TVB-N values observed in the ME (43.98 ± 2.81), CH (48.86 ± 1.26), and MECH (46.41 ± 2.33 mg N/100 g) groups may be attributed to the antimicrobial effects of muicle extract and chitosan, used alone or in combination as edible coatings. These treatments likely inhibited bacterial proliferation, reducing the microbial oxidative deamination of non-protein nitrogen compounds and thereby limiting TVB-N accumulation [67].

### 3.4. Microbiological Determinations

Figure 7 illustrates the behaviour of the aerobic mesophilic bacterial load in cazon fillets from the control group (C) and the treated groups (ME, CH, and MECH) over 18 days of ice storage. At the beginning of the storage period, no significant differences were observed between the control and the treatment groups (*p* > 0.05), with initial counts ranging from 3.86 ± 0.20 to 4.28 ± 0.08 Log CFU/g. These values are lower than those reported by Montaño-Cota et al. [56], who observed an initial microbial load of 5.3 ± 0.13 Log CFU/g in sierra (*Scomberomorus sierra*) muscle.

As storage time progressed, a significant increase in microbial load (*p* < 0.05) was recorded in both the control and treatment groups. The control group (C) reached a final count of 6.89 ± 1.82 Log CFU/g. Aref et al. [68] suggested that such increases in microbial load may result from the accumulation of simple nitrogenous compounds (e.g., amino acids and nucleotides) and free fatty acids, which are produced by the enzymatic degradation of fats and proteins inherent to the muscle tissue, creating favourable conditions for bacterial proliferation.

The final microbial loads for the treated groups were 6.54 ± 0.09 Log CFU/g (ME), 6.47 ± 0.20 Log CFU/g (CH), and 6.48 ± 0.17 Log CFU/g (MECH). Notably, none of the samples—including the control or treatment groups—exceeded the microbiological safety threshold of 7 Log CFU/g established by the International Commission on Microbiological Specifications for Foods (ICMSF) for fresh fish considered safe for human consumption [69]. Furthermore, the ME, CH, and MECH treatments exhibited significantly lower microbial counts compared to the control group (*p* < 0.05), confirming the antibacterial activity of muicle extract and chitosan, either individually or in combination, when applied as edible coatings.

These findings are consistent with the results obtained from pH and TVB-N analyses, where the ME, CH, and MECH treatments exhibited the lowest values. This supports the effectiveness of muicle extract and chitosan—either used individually or in combination as edible coatings—in inhibiting bacterial growth. These results highlight the potential of such formulations to delay microbial spoilage in cazon fillets stored on ice, thereby enhancing their quality and extending their shelf life.

## 4. Conclusions

The application of edible coatings based on muicle extract, either alone or combined with chitosan, contributed to preserving the physicochemical and microbiological quality of cazón fillets during ice storage. The muicle extract demonstrated antioxidant and antimicrobial activities, helping to control microbial growth and the accumulation of nitrogenous compounds (TVB-N) compared to untreated controls. The results of this study demonstrated that the aqueous extract of muicle exhibits significant antibacterial activity against *S. putrefaciens* and *L. monocytogenes*, confirming the presence of bioactive compounds capable of inhibiting bacterial growth and proliferation. These findings highlight the potential of muicle extract as a natural ingredient for extending the shelf life and improving the quality of fishery products.

## Figures and Tables

**Figure 1 foods-14-01619-f001:**
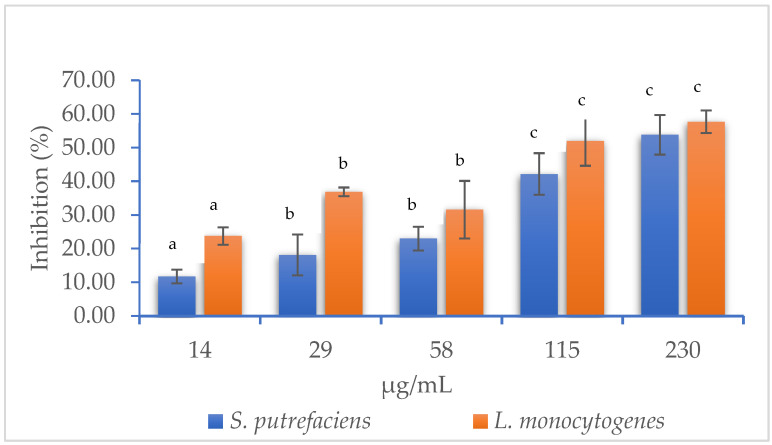
Inhibitory effect (%) of muicle aqueous extract at different concentrations against *S. putrefaciens* and *L. monocytogenes*. Bars sharing the same superscript letter within each bacterial species are not significantly different (*p* < 0.05).

**Figure 2 foods-14-01619-f002:**
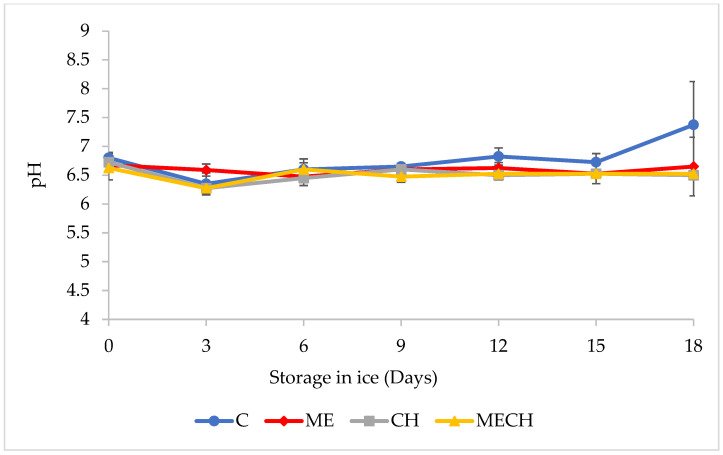
Changes in pH values of cazón fillets from the control group (C) and the ME, CH, and MECH treatments during 18 days of ice storage. Values are expressed as mean ± SD (n = 4).

**Figure 3 foods-14-01619-f003:**
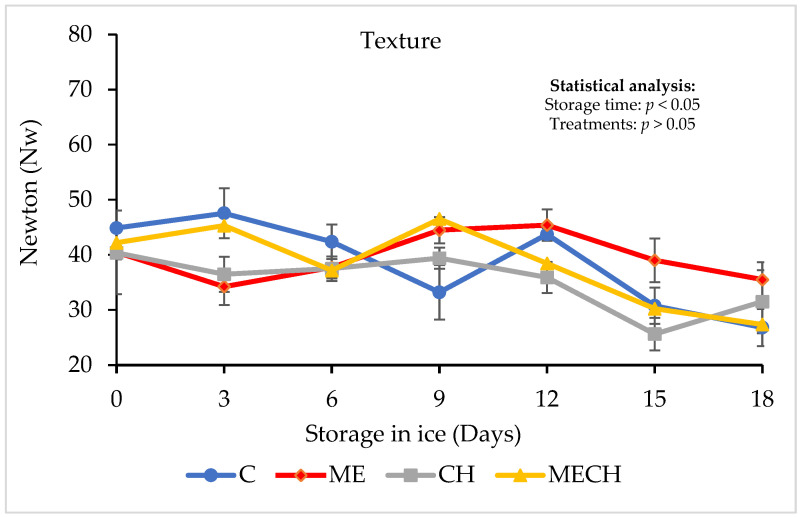
Changes in the texture (shear force) of cazon fillets from the control (C) and the ME, CH, and MECH treatments during 18 days of ice storage. Values are expressed as mean ± SD (n = 4).

**Figure 4 foods-14-01619-f004:**
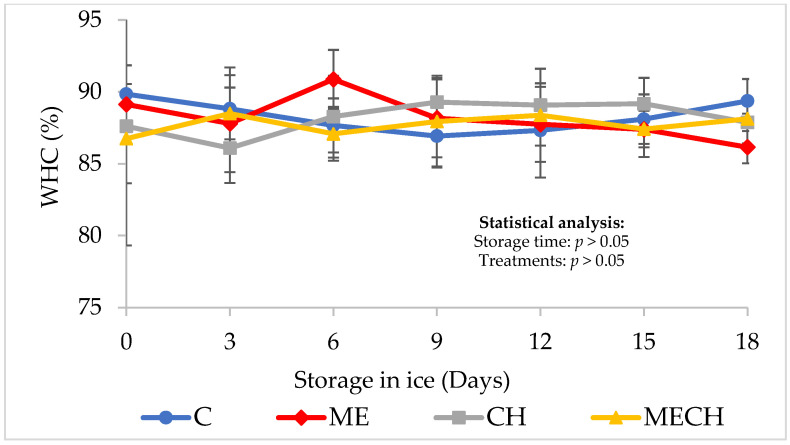
Changes in water-holding capacity (WHC) of cazón fillets the control (C) and the ME, CH, and MECH treatments during 18 days of ice storage. Values are expressed as mean ± SD (n = 4).

**Figure 5 foods-14-01619-f005:**
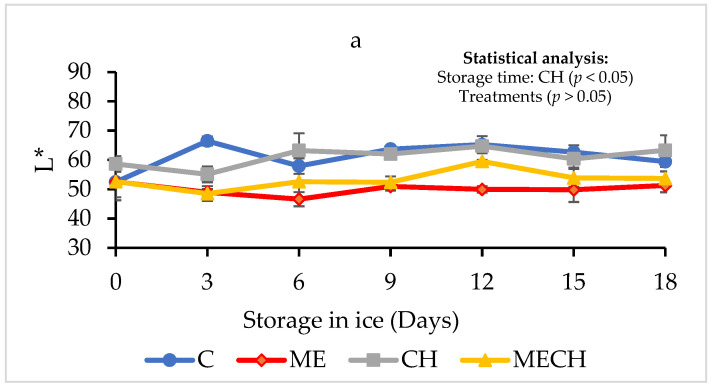
Changes in the colour parameters (L*, a*, and b*) of cazon fillets from the control (C) and the ME, CH, and MECH treatments during 18 days of ice storage. Figure (**a**) shows L*values, figure (**b**) shows a* values, and figure (**c**) shows b* values. Values are expressed as mean ± standard deviation (n = 4).

**Figure 6 foods-14-01619-f006:**
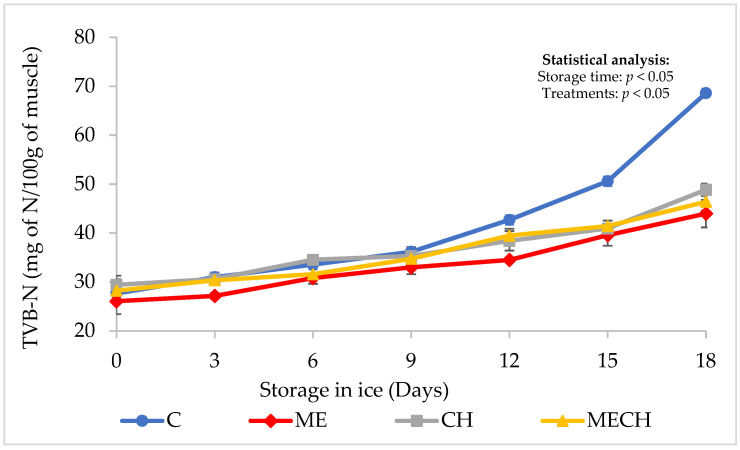
The total volatile basic nitrogen (TVB-N) content of cazon fillets from the control (C) and the ME, CH, and MECH treatment groups during 18 days of ice storage. Values are expressed as mean ± standard deviation (n = 4).

**Figure 7 foods-14-01619-f007:**
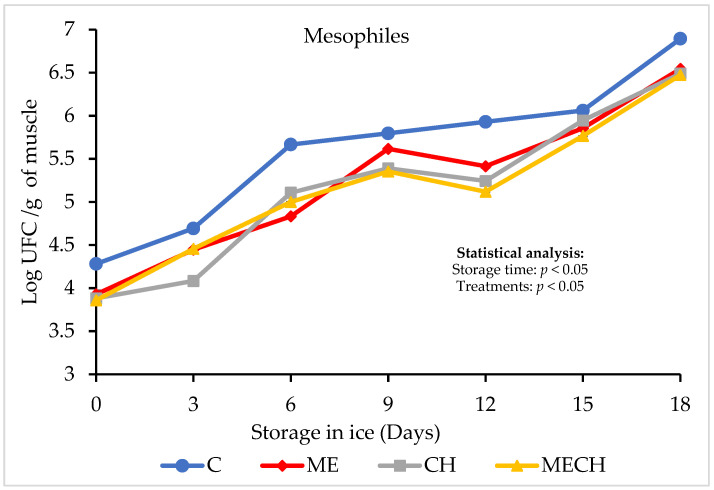
Aerobic mesophilic bacterial counts of cazon fillets from the control (C) and ME, CH, and MECH treatment groups during 18 days of ice storage. Values are expressed as mean ± standard deviation (n = 4).

**Table 1 foods-14-01619-t001:** Minimum inhibitory concentration (MIC, µg/mL) and IC_50_ values of aqueous *Justicia spicigera* extract against spoilage bacteria *S. putrefaciens* and *L. monocytogenes.* Data in the same line with different superscript letters are significantly different (*p* < 0.05).

	*S. putrefaciens*	*L. monocytogenes*
MIC (μg/mL ± SD)	3.01 ± 0.73 ^a^	0.09 ± 0.07 ^b^
IC50 (μg/mL ± SD)	204.56 ± 20.22 ^a^	118.09 ± 14.51 ^b^

**Table 2 foods-14-01619-t002:** The concentration of total phenolic compounds and flavonoids in the aqueous extract of muicle.

	230 μg/mL of Extract
Total phenols (mg GAE/100 g of sample)	41.48 ± 3.28
Flavonoids mg QE/100 g of sample	27.52 ± 1.11

## Data Availability

The original contributions presented in this study are included in the article; further inquiries can be directed to the corresponding authors.

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
