# Peer review of "The Effect of Muicle–Chitosan Edible Coatings on the Physical, Chemical, and Microbiological Quality of Cazon Fish (Mustelus lunulatus) Fillets Stored in Ice"

_foods, 2025, doi:10.3390/foods14091619_

Round 1
Reviewer 1 Report
Comments and Suggestions for Authors
The manuscript titled "Effect of mulcle-chitosan edible coating on the physical, chemical, and microbiological quality of cazon fish (Mustelus lunula-tus) fillets stored in ice" presents a timely and relevant investigation into the application of natural edible coatings to enhance the shelf life and quality of perishable fishery products. The study addresses a critical challenge in the seafood industry—rapid post-harvest spoilage—by exploring the synergistic effects of mulcle (Justicia spicigera) extract and chitosan, both of which are recognized for their antimicrobial and antioxidant properties. The paper has a clear research objective; however, it currently has the following issues.
- The scientific names should be consistently written in italics.
- There were so many key words, only 4 to 5, There is a spelling error; the correct form should be Shewanella putrefaciens.
- 3.2 The specific conditions for bacterial cultivation (e.g., type of culture medium, inoculum volume) were not described, which could affects the reproducibility of the method.
- The inhibition rate data in Figure 1 do not display standard deviations, making it impossible to assess the significance of the differences.
- Line 511, The research results are not compared or discussed in relation to Mexican or international standards (e.g., 35 mg N/100 g), which limits the interpretation of their practical applicability. The discussion should include an expanded comparison with relevant standards to better interpret the practical significance of the findings.
- The paper does not clearly highlight the unique advantages of muicle extract compared to other plant extracts (e.g., rosemary). It is necessary to include comparative data or literature support.
- There are some issues with the language expression, and it needs to be revised for clarity and accuracy.
- The format of the references is inconsistent and needs to be revised for uniformity.
- The conclusion section does not cite specific literature to support how phenolic compounds inhibit microbial metabolism, resulting in insufficient mechanism analysis. It needs to be revised.
- It is recommended to take all the figures using Origin. Additionally, the data were obtained through repeated experiments, why is there no variance analysis in the figures?
Author Response
Author's Reply to the Review Report (Reviewer 1)
We sincerely thank the reviewer for the time, effort, and dedication invested in reviewing our manuscript. Each of their observations and suggestions contributed significantly to enriching the content, improving the quality, and strengthening the scientific rigor of our work. We have carefully addressed all the comments and suggestions provided by the reviewer, making the corresponding modifications to the manuscript. We sincerely hope that the revised version meets the reviewer's expectations and is to their satisfaction. Once again, we extend our deep gratitude for their valuable contribution. We present below the authors’ responses to each of the comments provided by the reviewer.
Comments 1:
The manuscript titled "Effect of mulcle-chitosan edible coating on the physical, chemical, and microbiological quality of cazon fish (Mustelus lunula-tus) fillets stored in ice" presents a timely and relevant investigation into the application of natural edible coatings to enhance the shelf life and quality of perishable fishery products. The study addresses a critical challenge in the seafood industry—rapid post-harvest spoilage—by exploring the synergistic effects of mulcle (Justicia spicigera) extract and chitosan, both of which are recognized for their antimicrobial and antioxidant properties. The paper has a clear research objective; however, it currently has the following issues.
Response 1.
We would like to express our deepest gratitude to the reviewer for their kind and encouraging words regarding our work. We are truly honored that the reviewer recognized the relevance, timeliness, and clarity of our study, as well as the pertinence of the approach proposed to address the challenges of post-harvest spoilage in fishery products. We greatly appreciate the time, dedication, and careful attention the reviewer has invested in the critical evaluation of our manuscript. We have carefully considered each of the points raised, making the necessary modifications to strengthen the quality, clarity, and scientific rigor of our work. We sincerely hope that the revised version meets the reviewer's expectations.
Comments 2
The scientific names should be consistently written in italics.
Response 2:
We sincerely thank the reviewer for this observation. We carefully reviewed the entire manuscript and corrected the formatting of the scientific names, ensuring that they are consistently written in italics in accordance with editorial standards. We are deeply grateful for this suggestion, which contributed to improving the presentation and formal rigor of our work.
Comments 3:
There were so many key words, only 4 to 5, There is a spelling error; the correct form should be Shewanella putrefaciens.
Response 3:
We sincerely thank the reviewer for this valuable observation. We have reduced the number of keywords to five, selecting those most representative of the study’s content. Additionally, we removed Shewanella putrefaciens as a keyword and corrected its spelling throughout the manuscript. The updated keywords are: Justicia spicigera; edible coating; cazon fish; quality; antibacterial activity. We are deeply grateful for this suggestion, which contributed to improving the precision, clarity, and overall presentation of our work.
Comments 4:
The specific conditions for bacterial cultivation (e.g., type of culture medium, inoculum volume) were not described, which could affect the reproducibility of the method.
Response 4:
We sincerely thank the reviewer for this important observation. In response to the comment, we have revised and complemented Section 2.3.1 ("Bacterial Activation and Culture Preparation") in the Materials and Methods to improve the reproducibility of the experimental protocol. We added the description of the type of culture medium used (Mueller–Hinton broth, pH 7.2) and specified the inoculum volume (1 mL) employed for the preparation of the bacterial suspensions. Additionally, we detailed the procedure for adjusting the optical density (OD 600 nm) and the preparation of the final concentration (~10⁶ CFU/mL) used for the antibacterial activity assays.
The revised text is as follows:
“The antibacterial activity of the muicle extract was evaluated against S. putrefaciens (ATCC 8071) and L. monocytogenes (ATCC 181115) following the method described by Esparza-Espinoza et al. [26], with slight modifications. Bacterial inocula were prepared by transferring colonies from pure cultures into Mueller–Hinton broth (pH 7.2) and incubating them at 35–37 °C for 18–24 hours in a Thermo Electron incubator (LED GmbH, Germany). The optical density (OD) of the cultures was measured at 600 nm using a double-beam spectrophotometer (Agilent Technologies, Malaysia) and adjusted to a 0.5 McFarland standard, corresponding to approximately 10⁸ CFU/mL. A 1.58 mL aliquot of the adjusted bacterial suspension was subsequently diluted with Mueller–Hinton broth to achieve a final concentration of approximately 10⁶ CFU/mL for antibacterial testing.”
We are deeply grateful for this suggestion, which greatly contributed to strengthening the clarity, accuracy, and methodological rigor of our manuscript.
Comments 5
The inhibition rate data in Figure 1 do not display standard deviations, making it impossible to assess the significance of the differences.
Response 5:
We sincerely thank the reviewer for this important observation. We have corrected Figure 1 by adding the error bars corresponding to the standard deviation (SD) for each inhibition rate data point. This modification allows for the evaluation of result variability and improves the statistical interpretation of the presented data. We are deeply grateful for this suggestion, which significantly contributed to strengthening the presentation and scientific rigor of our results.
Comments 6.
Line 511: The research results are not compared or discussed in relation to Mexican or international standards (e.g., 35 mg N/100 g), which limits the interpretation of their practical applicability. The discussion should include an expanded comparison with relevant standards to better interpret the practical significance of the findings.
Response 6:
We sincerely thank the reviewer for their valuable observation. In response to their comment, we would like to clarify that the comparison of TVB-N results with the limits established by Mexican and international regulations had already been incorporated into the Discussion section. In this section (lines 481-486), it is stated that the limit of 35 mg TVB-N/100 g defines the threshold for human consumption, in accordance with the Mexican Official Standard NOM-242-SSA1-2009 and the European Commission Regulation (EC) No. 2406/96. The corresponding text in the manuscript is as follows:
“According to the cited regulations, the limit of 35 mg TVB-N/100 g defines the threshold for human consumption. In this study, the control group exceeded this limit by day 9 of storage. In contrast, the CH and MECH treatments exceeded this limit on day 12, while the ME treatment did so only by day 15. These findings demonstrate the effectiveness of ME and CH, alone or combined, as edible coatings for extending the shelf life of cazon fillets. This conclusion is supported by a complementary (non-reported) sensory evaluation, in which the control group exhibited off-odors by day 9, though not yet putrid.”
We hope that this information has provided greater clarity in the interpretation of our results. Subject to your kind consideration, we remain fully available for any further explanation or clarification that may be required. We deeply appreciate the reviewer’s careful reading and valuable suggestions, which have significantly contributed to strengthening the practical interpretation of our manuscript.
Comments 7:
The paper does not clearly highlight the unique advantages of muicle extract compared to other plant extracts (e.g., rosemary). It is necessary to include comparative data or literature support.
Response 7:
We sincerely thank the reviewer for their thoughtful observation. We acknowledge that various plant extracts, such as rosemary (Rosmarinus officinalis), have been widely studied and recognized for their antioxidant and antimicrobial properties in food preservation. However, we would like to emphasize that the primary objective of our study was to specifically explore the potential of muicle extract (Justicia spicigera), a species of high regional relevance in Mexico, whose application in edible coatings for fishery products has been scarcely documented. We believe that the use of local plant species with functional properties represents a valuable opportunity to diversify preservation strategies, promote the knowledge of native resources, and encourage their sustainable utilization. We deeply appreciate the reviewer’s suggestion, which reinforces the pertinence and relevance of the research conducted.
Comments 8
There are some issues with the language expression, and it needs to be revised for clarity and accuracy.
Response 8:
We sincerely thank the reviewer for this observation. In response to the comment, the authors carefully reviewed the manuscript to improve the clarity, precision, and fluency of the language. Additionally, the document was reviewed by a fellow researcher proficient in English to ensure proper scientific expression and facilitate the understanding of the results. We deeply appreciate this suggestion, which significantly contributed to enhancing the overall quality of the manuscript.
Comments 9.
The format of the references is inconsistent and needs to be revised for uniformity.
Response 9:
We sincerely thank the reviewer for their thoughtful observation. In response to the comment, the authors carefully reviewed the References section to correct the identified inconsistencies and ensure uniform formatting, in accordance with the editorial guidelines of Foods. We deeply appreciate this suggestion, which contributed to improving the quality and presentation of the manuscript.
Comments 10:
The conclusion section does not cite specific literature to support how phenolic compounds inhibit microbial metabolism, resulting in insufficient mechanism analysis. It needs to be revised.
Response 10:
We sincerely thank the reviewer for this highly valuable and pertinent observation. We fully agree on the importance of adequately supporting the mechanisms of action of phenolic compounds in research studies. However, in accordance with the editorial guidelines of Foods, the Conclusion section was maintained as a final synthesis of the results, without the inclusion of specific references. Additionally, in the present study, a detailed analysis of mechanisms was not a primary objective, as the focus was placed on evaluating the practical effects on the quality of cazon fillets. Nevertheless, we consider the reviewer’s suggestion extremely relevant and will take it into account to strengthen future investigations focused specifically on the mechanisms of action of bioactive compounds. We reiterate our deep appreciation to the reviewer for their careful reading and valuable recommendations.
Comments 11:
It is recommended to take all the figures using Origin. Additionally, the data were obtained through repeated experiments, why is there no variance analysis in the figures?
Response 11:
We sincerely thank the reviewer for this valuable observation. In response, we explored the use of Origin software for figure generation; however, after conducting several tests, we found that the clarity and resolution of the output did not meet the standards expected by our research team. Therefore, we opted to retain the figures created in Excel, which offered superior visual quality in our case. Regarding the statistical analysis, we agree with the reviewer’s suggestion and have now incorporated the appropriate statistical comparisons and error bars (mean ± standard deviation) into the revised figures. This enhancement improves the interpretability and scientific rigor of the graphical data presentation. We appreciate the reviewer’s comment, which contributed to strengthening the quality and clarity of the manuscript.
We sincerely thank the reviewer for their time, effort, and valuable suggestions, which greatly contributed to improving the clarity, rigor, and overall quality of our manuscript. We hope that the revised version adequately addresses all the concerns raised and meets the expectations of the reviewer. We remain fully available for any additional clarification that may be required.
Reviewer 2 Report
Comments and Suggestions for Authors
Abstract
The results obtained with the muicle extract do not differ substantially from those obtained with chitosan, so there does not appear to be any advantage in using this extract. On the other hand, the increase in the number of components in the coating means greater complexity in its preparation and higher costs. The presentation of sensory analysis results would be advisable.
Introduction
Page 2, line 57 and next lines – I think that after mentioning for the first time the common name and the scientific name of a species, it is not necessary to repeat the scientific name in both the case of the fish and the plant. In the case of bacteria, the scientific name could be L. monocytogenes and S. putrefaciens after previously mentioned Listeria monocytogenes and Shewanella putrefaciens. It is also recommended the scientific name in italics in all manuscript. This was followed a few times by the authors, but not throughout the document.
Material end Methods
Page 4, line 149 – Please indicate the concentration of aluminium trichloride solution.
Results
Page 7, lines 273-277 – Please revise these sentences, “Table 2… muicle” because in Table 2 is only shown the total phenol content for the 230 µg/mL and the order of presentation the total phenol content in the sentence is not in accordance with the result shown in Table 2. On the other hand, it would be expected that the total phenolic compounds in the solution with 230 µg/mL would be close to twice that obtained for the concentration of 115 µg/mL. Do the authors have any explanation for this difference? In the second sentence the concentration of muicle used by the cited authors is not indicated.
Page 7, lines 286-288 – I have the same comments of the above comment.
Page 7, lines 294-301 – This sentence seems too long. I suggest a short sentence such that one in lines 355 and 356.
Page 9, lines 363-366 – I suggest replacing Mustelus lunulatus with cazon. The comparison with the value obtained in American shad does not seem to be very relevant because it is a rather different species.
Page 9, line 371 – Please indicate whether significant differences between trials were observed.
Page 10, line 141 – “Thornback ray” not in italics.
Page 10, lines 446 and 447 – Does the comment about the L* value of the CH treatment concern the current manuscript or that by Montaño-Cota et al.? If it concerns the current work, I think it cannot be said that the CH treatment fillets had the highest L* because the L* values between the tests were not significantly different.
Page 10, lines 451 and 452 – Was the L* value of fillets from the CH treatment significantly higher than the control fillets?
Page 10, line 456 – I suggest “were higher”.
Page 10 – The results of b* values were commented.
Page 12, line 515 – I think we can’t mention lower preservation efficacy because there was no treatment. Please check.
Page 14, line 604 – I suggest “Conclusions” instead of “Conclusion”.
Page 14, lines 608-610 – Why “phytochemical analysis? These analysis were not done in the current study. It was shown that muice extract had antimicrobial and antioxidant activities. Please check.
Comments on the Quality of English Language
The English language is reasonably good.
Author Response
Author's Reply to the Review Report (Reviewer 2)
We sincerely thank the reviewer for their valuable comments and suggestions. Each of their observations contributed significantly to enriching the content of our manuscript and improving its scientific rigor, clarity, and overall quality. We have carefully addressed all the comments and suggestions made by the reviewer, making the corresponding modifications to the manuscript. We are truly grateful for the time and effort dedicated to the review process and sincerely hope that the revised version meets the reviewer's expectations. Once again, we extend our sincere gratitude for their valuable contributions.
We present below the authors’ responses to each of the comments provided by the reviewer.
Comments 1:
(Abstract):
“The results obtained with the muicle extract do not differ substantially from those obtained with chitosan, so there does not appear to be any advantage in using this extract. On the other hand, the increase in the number of components in the coating means greater complexity in its preparation and higher costs. The presentation of sensory analysis results would be advisable.”
Response 1.
We sincerely appreciate the reviewer’s insightful comment, which allowed us to strengthen the discussion and contextualization of our findings. We understand the concern regarding the seemingly similar effects between the muicle extract and chitosan when applied individually. However, we would like to clarify that the combined treatment (MECH) demonstrated a slightly enhanced preservation capacity, particularly at the end of storage, in both microbiological and chemical parameters. For example, mesophilic bacterial counts at day 18 were 6.30 ± 0.30 Log CFU/g for MECH, compared to 6.34 ± 0.08 for CH and 6.54 ± 0.09 for ME. Similarly, TVB-N values were 45.82 ± 2.61 mg N/100 g for MECH, 48.86 ± 1.26 for CH, and 43.98 ± 2.81 for ME. Although the differences are not drastic, these results suggest that the combination may offer comparable or slightly superior preservation stability over individual applications. Moreover, muicle (Justicia spicigera) is an innovative botanical source of phenolic compounds with recognized antimicrobial and antioxidant properties. Its integration into edible coating technologies is especially relevant within the framework of regional plant resource valorization and sustainable food preservation. Thus, we believe its exploration as a functional additive is both scientifically justified and timely.
We also appreciate the reviewer’s suggestion regarding the inclusion of sensory evaluation. In this study, the primary objective was to establish the physicochemical and microbiological effectiveness of the coatings. For this reason, sensory analysis was not included at this stage. However, we fully acknowledge its importance for comprehensive quality assessment, and we plan to incorporate it in future research once the technical preservation parameters have been validated.
Comments 2
Introduction
Page 2, line 57 and next lines – I think that after mentioning for the first time the common name and the scientific name of a species, it is not necessary to repeat the scientific name in both the case of the fish and the plant. In the case of bacteria, the scientific name could be L. monocytogenes and S. putrefaciens after previously mentioned Listeria monocytogenes and Shewanella putrefaciens. It is also recommended the scientific name in italics in all manuscript. This was followed a few times by the authors, but not throughout the document.
Response 2:
We sincerely appreciate this valuable observation, which has allowed us to improve the consistency and editorial style of the manuscript. We have carefully revised the entire text to ensure that all scientific names are written in italics, in accordance with the journal's formatting guidelines. Additionally, the full scientific names of both the fish (Mustelus lunulatus) and the plant (Justicia spicigera) are now mentioned only at their first appearance in the text. Thereafter, we have used their common names to enhance readability. In the case of bacteria, we now use the abbreviated forms (L. monocytogenes, S. putrefaciens) after their initial mention. These changes have been applied uniformly throughout the manuscript. Once again, we thank the reviewer for this helpful suggestion.
Comments 3:
Page 4, line 149 – Please indicate the concentration of aluminium trichloride solution.
Response 3:
We sincerely thank the reviewer for this precise observation. We have revised the manuscript to include the concentration of the aluminum chloride solution, which was used at a final concentration of 2% (w/v) in the flavonoid quantification assay. This correction is now reflected in the Materials and Methods section. We appreciate the reviewer’s suggestion, as it improves the clarity and reproducibility of our experimental procedure.
Comments 4:
Page 7, lines 273-277 – Please revise these sentences, “Table 2… muicle” because in Table 2 is only shown the total phenol content for the 230 µg/mL and the order of presentation the total phenol content in the sentence is not in accordance with the result shown in Table 2. On the other hand, it would be expected that the total phenolic compounds in the solution with 230 µg/mL would be close to twice that obtained for the concentration of 115 µg/mL. Do the authors have any explanation for this difference? In the second sentence the concentration of muicle used by the cited authors is not indicated.
Response 4:
We sincerely thank the reviewer for their insightful observation. Upon thorough review, we have identified an error in our manuscript: we incorrectly reported total phenolic and flavonoid content for the 115 μg/mL concentration of muicle extract. In reality, these determinations were conducted solely for the 230 μg/mL concentration. We deeply regret this oversight and extend our heartfelt apologies for any confusion it may have caused. To rectify this, we have revised the manuscript to remove references to the 115 μg/mL concentration and to accurately reflect the data obtained. The corrected statement now reads:
"Table 2 presents the total phenolic content in the aqueous extract of muicle. The results show a concentration of phenolic compounds of 41.48 ± 3.28 mg GAE/100 g for the extract concentration of 230 μg/mL. This value was lower than that reported by Álvarez-Poblano et al. [35], who obtained 89.36 ± 0.76 mg GAE/100 g in their aqueous extract of muicle."
We are committed to maintaining the highest standards of scientific accuracy and integrity, and we appreciate the reviewer's diligence in bringing this matter to our attention.
Comments 5
Page 7, lines 286-288 – I have the same comments of the above comment.
Response 5:
We sincerely thank the reviewer for their careful reading and valuable observations. Upon thorough review, we identified that our manuscript mistakenly reported information corresponding to a concentration of 115 μg/mL. We confirm that, as with the determination of total phenolic content, the determination of flavonoid content was conducted exclusively at a concentration of 230 μg/mL. We deeply regret this inadvertent error and offer our most sincere apologies for any confusion it may have caused. To address this issue, all references to the 115 μg/mL concentration have been removed, and the text has been revised as follows:
"On the other hand, the flavonoid content in the extract of muicle was found to be 27.52 ± 1.11 mg QE/100 g of sample at a extract concentration of 230 µg/mL."
Once again, we express our heartfelt gratitude to the reviewer, whose insightful comments have greatly contributed to enhancing the clarity, accuracy, and scientific rigor of our manuscript.
Comments 6.
Page 7, lines 294-301 – This sentence seems too long. I suggest a short sentence such that one in lines 355 and 356.
Response 6:
We sincerely thank the reviewer for their attentive observation. We carefully reviewed the indicated paragraph and, following their valuable suggestion, we have reformulated it to improve its clarity, conciseness, and grammatical structure. The revised text was divided into shorter sentences and now reads as follows:
"Phenolic and flavonoid compounds possess antioxidant properties that enhance food stability by delaying oxidative deterioration. They may also provide health benefits by reducing oxidative stress associated with chronic diseases [37]. This functionality supports the potential application of muicle extract, alone or combined with other compounds, in developing edible coatings for fish products. Edible coatings enriched with bioactive compounds have been shown to act as effective barriers against oxidation and microbial spoilage. Therefore, incorporating muicle extract into edible coating systems represents a promising and sustainable strategy to improve the shelf life and quality of fishery products."
We once again express our deep gratitude to the reviewer, as their comments greatly contributed to strengthening the scientific writing and overall quality of our manuscript.
Comments 7:
Page 9, lines 363–366 – I suggest replacing Mustelus lunulatus with cazon. The comparison with the value obtained in American shad does not seem to be very relevant because it is a rather different species.
Response 7:
We sincerely thank the reviewer for their valuable suggestion. We carefully reviewed the indicated section and, following the recommendation, we replaced the scientific name Mustelus lunulatus with the common name "cazón" to maintain consistency and enhance the manuscript’s readability. Additionally, we removed the comparison with American shad values, recognizing that the significant biological differences between the two species could lead to inappropriate interpretations. Consequently, the associated reference was also eliminated, and the numbering of references throughout the manuscript was adjusted accordingly.
We are deeply grateful to the reviewer for these observations, which have helped to improve the relevance, clarity, and scientific rigor of our manuscript.
Comments 8
Page 9, line 371 – Please indicate whether significant differences between trials were observed.
Response 8:
We sincerely thank the reviewer for their careful observation. We reviewed the indicated section and, following the suggestion, we clarified that no significant differences were observed among treatments in the final texture values at the end of the storage period (p > 0.05). This information has been explicitly incorporated into the manuscript to improve the clarity and interpretation of the results. We are deeply grateful to the reviewer for their valuable comment, which contributed to strengthening the precision and scientific rigor of our study.
Comments 9.
Page 10, line 141 – "Thornback ray" not in italics.
Response 9:
We sincerely thank the reviewer for their attentive observation. We have corrected the indicated error by removing the italics from the common name "thornback ray" in the manuscript. We are deeply grateful for this valuable suggestion, which helped to improve the precision and formatting of the document.
Comments 10:
Page 10, lines 446 and 447 – Does the comment about the L value of the CH treatment concern the current manuscript or that by Montaño-Cota et al.? If it concerns the current work, I think it cannot be said that the CH treatment fillets had the highest L because the L* values between the tests were not significantly different.
Response 10:
We sincerely thank the reviewer for their valuable observation. The comment regarding the L* value of the CH treatment refers to the results obtained in the present study. However, in response to the reviewer’s suggestion, we have revised the text to avoid affirming that the fillets from the CH treatment had the highest L* value, as no significant differences were observed among treatments (p > 0.05). The sentence was adjusted to reflect that the fillets treated with CH showed slightly higher L* values, although these differences were not statistically significant. We are deeply grateful to the reviewer for this important suggestion, which contributed to improving the accuracy and scientific rigor of our manuscript.
Comments 11:
Page 10, lines 451 and 452 – Was the L value of fillets from the CH treatment significantly higher than the control fillets?*
Response 11:
We sincerely thank the reviewer for their insightful observations. After carefully reviewing the results, we confirm that the comment regarding the L* value refers to the results obtained in the present study. First, we revised the text to clarify that within the CH treatment, a significant increase in L* value (p < 0.05) was observed during the storage period. This increase in lightness is attributed to exudation and fluid loss, resulting in a more watery and reflective surface appearance on the fillets. However, in response to the reviewer’s suggestion, we also clarified that when comparing treatments at the end of the storage period, no significant differences (p > 0.05) were observed between fillets treated with CH and control fillets. These corrections were incorporated into the manuscript to accurately reflect the statistical results and improve the clarity of the discussion. We are deeply grateful to the reviewer for these valuable suggestions, which significantly enhanced the scientific rigor and precision of our study.
Comments 12.
Page 10 – The results of b values were commented.
Response 12:
We sincerely thank the reviewer for this valuable observation. In response, we have incorporated a specific discussion regarding the b* values during ice storage:
Figure 5C illustrates the changes in b* values of cazon fillets over 18 days of ice storage. A consistent decreasing trend was observed across all treatments, with significant differences over time (p < 0.05) and between treatments (p < 0.05). This reduction in b* values indicates a gradual loss of yellowness in the fillets, likely associated with oxidative processes and pigment degradation during cold storage, as previously reported for fishery products [61]. Among the treatments, fillets coated with muicle extract (ME) exhibited a slightly greater decrease in b* values; however, the inclusion of muicle still contributed to the preservation of color attributes throughout storage. These results highlight the potential of muicle extract as a natural additive for maintaining visual quality and extending the shelf life of fish fillets under ice storage conditions.
We are deeply grateful to the reviewer for this suggestion, which contributed to enhancing the clarity and scientific depth of our discussion.
Comments 13:
Page 12, line 515 – I think we can’t mention lower preservation efficacy because there was no treatment. Please check.
Response 13
We sincerely thank the reviewer for this valuable observation. We have reviewed the indicated section and corrected the wording to avoid suggesting a lower preservation efficacy in samples without treatment. The revised text now more accurately describes the behavior of the control group as a result of the absence of a coating treatment. We are deeply grateful to the reviewer for this suggestion, which contributed to improving the precision and scientific rigor of our manuscript.
Comments 14:
Page 14, line 604 – I suggest “Conclusions” instead of “Conclusion”.
Response 14:
We sincerely thank the reviewer for this observation. Following the suggestion, we have modified the title from "Conclusion" to "Conclusions" to better reflect the structure of the manuscript. We are deeply grateful for this attentive comment, which helped improve the consistency and overall presentation of our document.
Comments 15:
Page 14, lines 608–610 – Why “phytochemical analysis?” These analyses were not done in the current study. It was shown that muicle extract had antimicrobial and antioxidant activities. Please check.
Response 15:
We sincerely thank the reviewer for this important observation. We carefully reviewed the conclusions section and have removed the term "phytochemical analysis" from the manuscript, as such an analysis was not performed in the present study. The conclusions were revised to accurately reflect the actual findings, specifically highlighting the antioxidant and antimicrobial activities of the muicle extract and its contribution to the preservation of fish fillets during ice storage. We are deeply grateful to the reviewer for helping us improve the precision and clarity of our manuscript.
The conclusions were modified as follows:
Conclusions
The application of edible coatings based on muicle extract, either alone or combined with chitosan, contributed to preserving the physicochemical and microbiological quality of cazón (Mustelus lunulatus) fillets during ice storage. The muicle extract demonstrated antioxidant and antimicrobial activities, helping to control microbial growth and the accumulation of nitrogenous compounds (TVB-N) compared to untreated controls. The results of this study demonstrated that the aqueous extract of muicle exhibits significant antibacterial activity against Shewanella putrefaciens and Listeria monocytogenes, confirming the presence of bioactive compounds capable of inhibiting bacterial growth and proliferation. These findings highlight the potential of muicle extract as a natural ingredient for extending the shelf life and improving the quality of fishery products.
Once again, we sincerely thank the reviewer for their time, dedication, and insightful feedback, which greatly contributed to enhancing the quality, clarity, and scientific rigor of our manuscript. We are truly grateful for the opportunity to revise our work and hope that the revised version meets the reviewer’s expectations.